# Prognostic Factors Associated with Breast Cancer-Specific Survival from 1995 to 2022: A Systematic Review and Meta-Analysis of 1,386,663 Cases from 30 Countries

**DOI:** 10.3390/diseases12060111

**Published:** 2024-05-23

**Authors:** Hanif Abdul Rahman, Siti Nurzaimah Nazhirah Zaim, Ummi Salwa Suhaimei, Al Amin Jamain

**Affiliations:** PAPRSB Institute of Health Sciences, Universiti Brunei Darussalam, Tungku Link Road, Gadong BE1410, Brunei; 21m8731@ubd.edu.bn (S.N.N.Z.); 23h1801@ubd.edu.bn (U.S.S.); 23h1804@ubd.edu.bn (A.A.J.)

**Keywords:** breast cancer specific survival, prognostic factor, hazard ratio, systematic review, meta-analysis

## Abstract

Breast cancer is the fifth-ranked cancer globally. Despite early diagnosis and advances in treatment, breast cancer mortality is increasing. This meta-analysis aims to examine all possible prognostic factors that improve/deteriorate breast cancer-specific survival. MEDLINE, PubMed, ScienceDirect, Ovid, and Google Scholar were systematically searched until September 16, 2023. The retrieved studies from 1995 to 2022 accumulated 1,386,663 cases from 30 countries. A total of 13 out of 22 prognostic factors were significantly associated with breast cancer-specific survival. A random-effects model provided a pooled estimate of the top five poorest prognostic factors, including Stage 4 (HR = 12.12; 95% CI: 5.70, 25.76), followed by Stage 3 (HR = 3.42, 95% CI: 2.51, 4.67), a comorbidity index ≥ 3 (HR = 3.29; 95% CI: 4.52, 7.35), the poor differentiation of cancer cell histology (HR = 2.43; 95% CI: 1.79, 3.30), and undifferentiated cancer cell histology (HR = 2.24; 95% CI: 1.66, 3.01). Other survival-reducing factors include positive nodes, age, race, HER2-receptor positivity, and overweight/obesity. The top five best prognostic factors include different types of mastectomies and breast-conserving therapies (HR = 0.56; 95% CI: 0.44, 0.70), medullary histology (HR = 0.62; 95% CI: 0.53, 0.72), higher education (HR = 0.72; 95% CI: 0.68, 0.77), and a positive estrogen receptor status (HR = 0.78; 95% CI: 0.65, 0.94). Heterogeneity was observed in most studies. Data from developing countries are still scarce.

## 1. Introduction

According to the Global Burden of Disease Cancer study, breast cancer remains the fifth-ranked cancer globally, with an increase from 2005 to 2015 of 17.2% (95% CI: 9.3%, 24.3%) in absolute years of life lost (A-YLLs) [1] and a significant increase in total YLLs [2]. The number of breast cancer cases has generally increased in recent years due to population growth, ageing populations, and age-specific cases [3]. Despite increases in the number of breast cancer survivors due to early diagnosis and advances in treatment [4] breast cancer mortality rate has increased by 21.3% (95% CI: 14.9%, 27.2%) [2].

Survival rate is one of the main outcome measures to predict the effectiveness of treatment or intervention for a specific period after diagnosis [5,6]. The accuracy of breast cancer survival prediction models requires the identification of the most salient factors that establish risk factors, such as age, race, stage at diagnosis, tumour size, hormonal receptor status, type of treatment, and family history of breast cancer, which could provide some evidence [5]. However, over-simplified or parsimonious models are often poor when applied to future trends and, thus, a comprehensive review of other biological and non-biological factors providing a more realistic interplay of this complex relationship needs to be considered. Furthermore, the availability of various cancer survival statistics, such as all-cause mortality, cancer-specific mortality, crude probability, and relative survival rate, adds to the difficulty of having a constant unit of measurement across studies [7].

Based on the above and a lack of systematic review and meta-analysis that comprehensively covers major factors related to breast cancer-specific survival, we conducted a systematic review and meta-analysis of longitudinal observational studies that report breast cancer-specific survival using a Hazard Ratio and a 95% confidence interval according to the preferred items for reporting systematic review and meta-analysis (PRISMA) guidelines [8].

## 2. Materials and Methods

### 2.1. Eligibility Criteria

Studies with the following eligibility characteristics were included: (1) they employed a longitudinal (retrospective or prospective) design, (2) they examined the survival of breast cancer patients using breast cancer-specific mortality, (3) a minimum sample size of 100, and (4) they provided a measure of survival statistics, particularly a Hazard Ratio (HR) and the corresponding 95% confidence interval (CI) for the estimation. Whenever necessary, the authors were contacted to provide more information to calculate these statistics. Studies that report survival statistics other than breast cancer-specific survival were excluded.

### 2.2. Search Strategy

A systematic search was performed using electronic databases, including (1) MEDLINE, (2) PubMed, (3) ScienceDirect, (4) Ovid, and (5) Google Scholar, spanning from 1990 to 2023. The search strategy was a combination of keywords that consisted of (“breast cancer” OR “breast carcinoma” OR “breast neoplasm” OR “tumor breast”) AND (“mortality” OR “survival”) AND (“factor” OR “prognostic factor”) AND (“Hazard ratio” OR “Cox model” OR “proportional hazard model”). The reference list of selected studies in the present review was also hand-searched in order to retrieve any additional relevant articles. Other non-primary sources such as editorials, conference proceedings, and reviews were excluded from the search.

### 2.3. Study Selection

The results of the systematic search were entered into a reference manager software (Mendeley v2.115.0), and two reviewers (HAR and SNNZ) independently screened the study titles, abstracts, and full texts based on the eligibility criteria. If there was disagreement, it was resolved through consensus with a third party.

### 2.4. Quality Assessment

The Quality in Prognosis Studies (QUIPS) tool was used to assess the risk of bias in the selected studies and consisted of the following items: (1) study participation, (2) study attrition, (3) prognostic factor measurement, (4) outcome measurement, (5) study confounding, and (6) statistical analysis and reporting. Two reviewers (HAR and SNNZ) assessed the quality of the retrieved articles independently. The inter-rater agreement using weighted kappa was used to obtain proper agreement between the two reviewers where values were tiered into 40–59% (low quality), 60–79% (moderate quality), and more than 90% (high quality). Only studies that were rated moderate and high were included in the review and meta-analysis.

### 2.5. Data Extraction

The data were extracted by two reviewers (HAR and SNNZ) independently using standard pre-defined features and formatting, including the first author’s name, the year of publication, the country of study, the study design/database used, the sample size/number of study participants, study population/sample characteristics, and prognostic factors studies in association with breast cancer-specific survival.

### 2.6. Statistical Analyses

To estimate the pooled effects of breast cancer-specific survival in association with each prognostic factor, the Hazard Ratios and corresponding 95% confidence intervals were combined and reported as fixed-effects models or random-effects models when heterogeneity is present. The pooled effect is considered statistically significant if the *p*-value is less than 0.05. The weight of each study calculated based on the inverse of the standard error is reported in percentages where a higher percentage indicates a higher weight. A forest plot is used to visualise the output of individual studies and pooled estimates where the diamond in the bottom represents the pooled effect size.

To explore between-study heterogeneity, the I-square (I^2^) statistics based on Cochran’s Q (following a Chi-square distribution), were used. I^2^ statistics above 50% are considered substantial [9]. Publication bias was assessed visually using funnel plots and Egger’s test to investigate the asymmetry among the study estimates. Microsoft Excel was used to input and prepare the data for analysis. All statistical analyses were performed in RStudio (v1.4.1717) using the *meta* [10] and *metafor* [11] packages.

## 3. Results

### 3.1. Study Characteristics

Figure 1 depicts the flow diagram of the systematic search process and study selection results. The initial search yielded 13,817 studies. After the exclusion of irrelevant studies, duplication, and other reasons, 1888 were screened based on title and abstract. Further exclusion based on eligibility criteria resulted in 60 studies included in this review, of which 33 (55%) were of moderate quality and 27 (45%) were of high quality (Table 1). Forty-one studies were eligible for meta-analysis.

Table 2 shows the individual characteristics of selected studies [12,13,14,15,16,17,18,19,20,21,22,23,24,25,26,27,28,29,30,31,32,33,34,35,36,37,38,39,40,41,42,43,44,45,46,47,48,49,50,51,52,53,54,55,56,57,58,59,60,61,62,63,64,65,66,67,68]. Overall, the selected studies originated from 30 countries, where the majority of the cases were from North America (Figure 2). Even though the search strategy was inclusive of 1990 to 2023, the eligible studies range between 1995 to 2022 (Figure 3).

### 3.2. Pooled Effects of Breast Cancer-Specific Survival

Table 3 presents the pooled estimate of breast cancer-specific survival for each prognostic factor. Following the meta-analysis procedure, we discovered several prognostic factors that are significantly associated with increased or decreased breast cancer survival. The top five poorest prognostic factors were Stage 4 cancer (HR = 12.12; 95% CI: 5.70, 25.76), followed by Stage 3 cancer (HR = 3.42, 95% CI: 2.51, 4.67), comorbidity index ≥ 3 (HR = 3.29; 95% CI: 4.52, 7.35), poor differentiation of cancer cell histology (HR = 2.43; 95% CI: 1.79, 3.30), and undifferentiated cancer cell histology (HR = 2.24; 95% CI: 1.66, 3.01). Other survival-reducing factors include positive nodes, age, race, Human Epidermal Growth factor receptor-2 (HER2) positivity, and body mass index. On the other hand, the top five prognostic factors that improve survival were surgery, including different types of mastectomies and breast-conserving therapies (HR = 0.56; 95% CI: 0.44, 0.70); medullary histology (HR = 0.62; 95% CI: 0.53, 0.72); higher education (HR = 0.72; 95% CI: 0.68, 0.77); positive estrogen receptor status (HR = 0.78; 95% CI: 0.65, 0.94); and secondary-level education (HR = 0.84; 95% CI: 0.79, 0.90). Figure 4, Figure 5, Figure 6, Figure 7, Figure 8, Figure 9, Figure 10, Figure 11, Figure 12 and Figure 13 illustrate the forest plots of breast cancer-specific survival for the top five survival-reducing and survival-improving factors. The forest plots of the remaining factors can be viewed in Appendix A.

### 3.3. Heterogeneity

The result from the Q and I^2^ statistics indicated considerable between-study heterogeneity in all prognostic factors except for age above 60 (I^2^ = 0%), higher education (I^2^ = 0%), medullary histology (I^2^ = 0%), comorbidity index 1 to 2 (I^2^ = 21%), oral contraceptive use (I^2^ = 0%), light to moderate physical activity (I^2^ = 0%), and high to vigorous physical activity (I^2^ = 18%) (Table 4).

### 3.4. Publication Bias

Funnel plots of each prognostic factor are presented in Appendix A. Egger’s test indicated that publication bias is present in ages 35 to 60, Stage 3, undifferentiated cancer cells, tumour size, overweight/obese, chemotherapy, and radiotherapy (Table 5).

### 3.5. Meta-Regression Analysis

Further exploring the source of heterogeneity, the univariable meta-regression analysis presented in Table 6, showed that there was a significant increase in studies of the above-60 age group (ß = 0.05, *p* = 0.002) and a significant decrease in studies of the HER2 receptor-negative group (ß = −0.05, *p* = 0.036) in increasing years of study. No significant change was observed in the sample size or study design.

## 4. Discussion

In the present systematic review and meta-analysis, we summarised the evidence for breast cancer-specific survival and discovered 13 out of 22 significant prognostic factors including surgery, estrogen receptor status, education, histology, body mass index, HER2 receptor status, race, tumour size, tumour differentiation, age, grade, node affected, comorbidity index, and cancer staging. The results showed that the top factor for improving breast cancer-specific survival was the surgical resection of the primary tumour using different types of mastectomies and breast-conserving therapies, with an increased survival rate of 44%. This reiterated the benefits of surgery by reducing the number of circulating tumour cells and improving disease outcomes, not just for breast cancer, but also potentially for other cancer types [69]. Nevertheless, the studies included in this meta-analysis of surgery were mostly from developed countries such as Canada, the USA, and the Nordic countries, and a clear evidence gap still exists for developing countries. In this meta-analysis, education status is also highlighted, where an increase in one’s level of education leads to an increase in breast cancer-specific survival by 28%. Women with a higher level of education generally have better uptake in breast cancer screening and, therefore, a better likelihood of early detection that results in higher breast cancer incidence and better survival outcome [70].

At the other extreme, breast cancer-specific survival was poorest in those with Stages 3 and 4 advanced cancer, which decreases survival rates by 3 and 12 times, respectively. This result further emphasises the importance of the early detection and diagnosis of breast cancer, and the implementation of screening programmes to provide timely treatment [71]. In this meta-analysis, the results also highlighted the important role of the comorbidity index. A higher score on the comorbidity index is significantly associated with higher mortality by more than three times. The current treatment guidelines for breast cancer have limited recommendations for comorbidities in decision-making, and no guidance to tailor treatments based on comorbidities [72]. This is mainly due to the challenges of obtaining evidence of drug efficacy for patients with comorbidities in clinical trials, which often exclude patients with comorbidities [73].

Other prognostic factors had significant associations but were not as strong as above. In addition, evidence of heterogeneity was observed in most studies and publication bias is present in certain factors including ages 35 to 60, Stage 3 cancer, undifferentiated cancer cells, tumour size, overweight/obese, chemotherapy, and radiotherapy. The source of heterogeneity was explored and did not yield significance. We postulated that due to the large sample size of observational studies selected in this meta-analysis, this would result in higher-power-to-detect, even clinically unimportant, heterogeneity [74].

Limitations of this study need to be considered when interpreting the results. The primary aim of this meta-analysis was to estimate all possible prognostic factors on breast cancer-specific survival; however, due to the restricted number of studies on certain factors, only 22 factors were analysed. A large number of studies originated from developed countries, and caution is warranted when applying the results to developing countries, where evidence is still lacking. Some prognostic factors have a relatively small number of studies. The survival rates of this study are as reported in each study and not distinguished, so they may be 1-, 3-, 5-, or more than 10-year survivals. Regardless of these limitations, this meta-analysis generated conclusive evidence for estimating the top five poorest and best prognostic factors for breast cancer-specific survival.

In conclusion, this meta-analysis estimated the pooled effects of breast cancer-specific survival in a large sample originating from 30 countries. The results highlight the beneficial effects of surgery, higher education, early detection, and the consideration of comorbidities in the treatment of breast cancer patients.

## Figures and Tables

**Figure 1 diseases-12-00111-f001:**
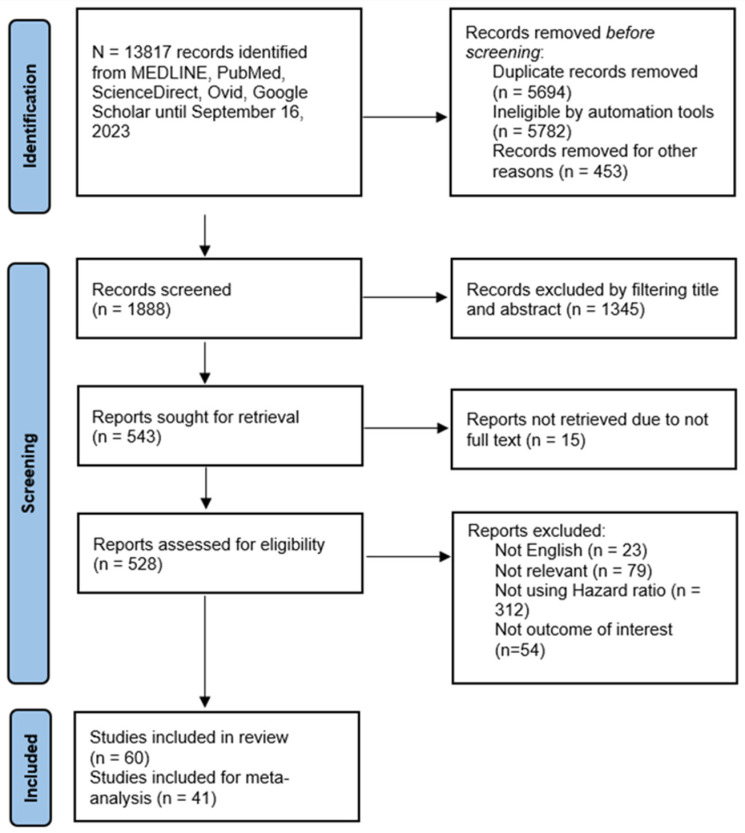
PRISMA flowchart of systematic search and study selection outcome.

**Figure 2 diseases-12-00111-f002:**
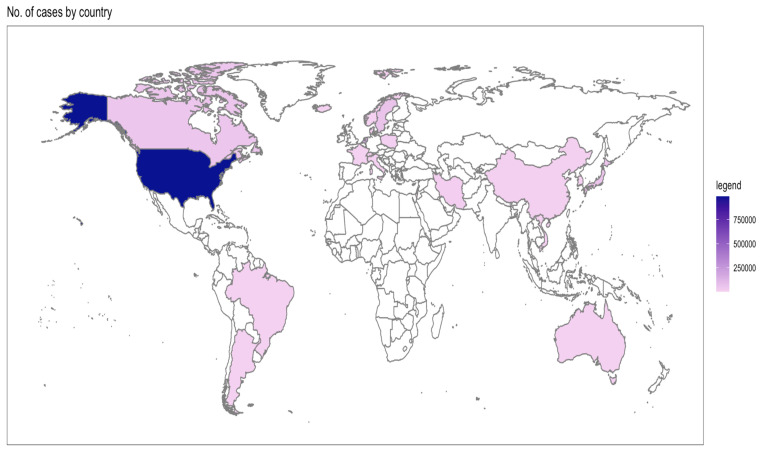
Distribution of selected studies by number of cases and country.

**Figure 3 diseases-12-00111-f003:**
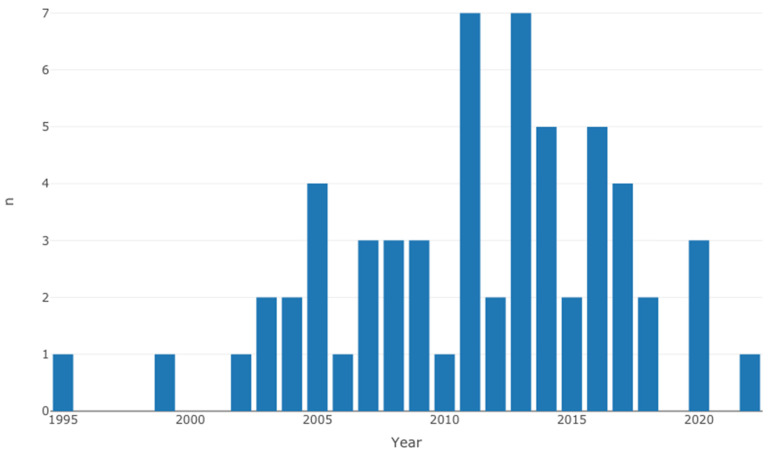
Distribution of selected studies by year of publication.

**Figure 4 diseases-12-00111-f004:**
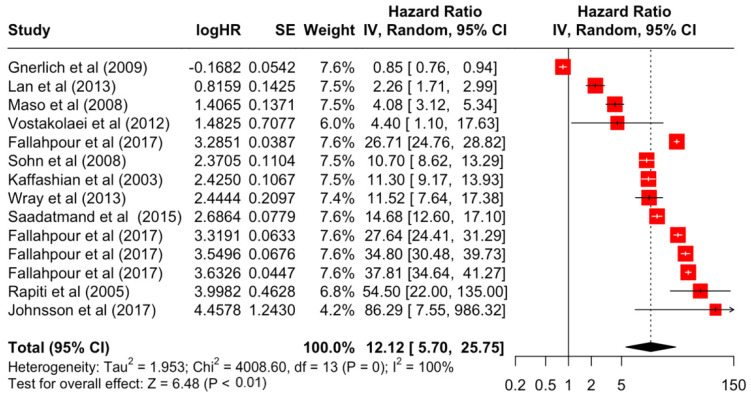
Forest Plot for Random-effects Hazard Ratio Model of Stage 4.

**Figure 5 diseases-12-00111-f005:**
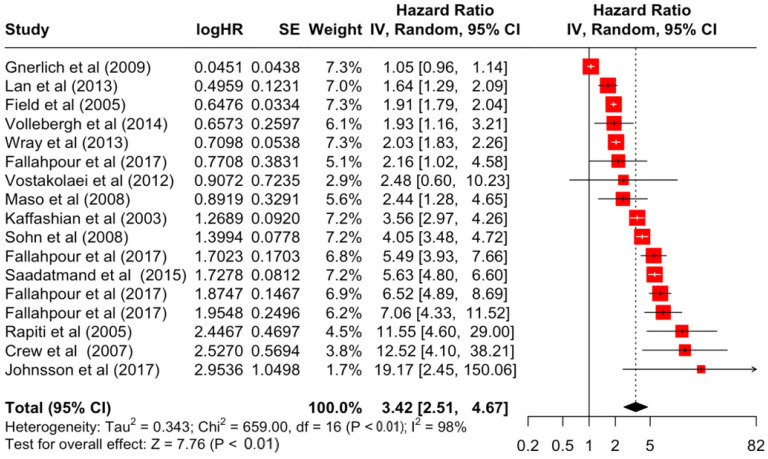
Forest Plot for Random-effects Hazard Ratio Model of Stage 3.

**Figure 6 diseases-12-00111-f006:**
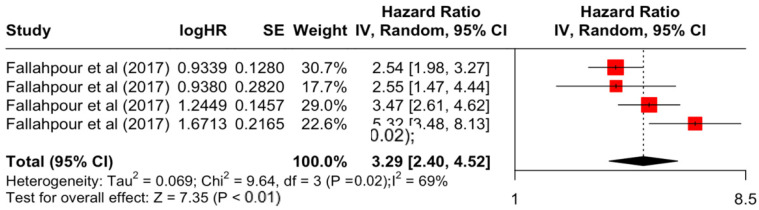
Forest Plot for Random-effects Hazard Ratio Model of Comorbidity Index (≥3).

**Figure 7 diseases-12-00111-f007:**
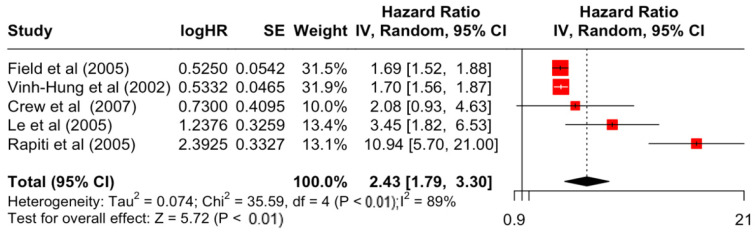
Forest Plot for Random-effects Hazard Ratio Model of Differentiation (Poor).

**Figure 8 diseases-12-00111-f008:**
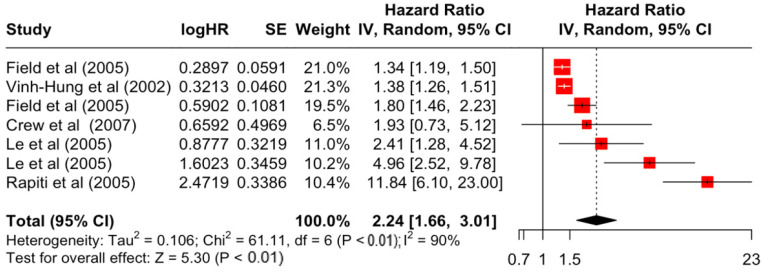
Forest Plot for Random-effects Hazard Ratio Model of Differentiation (Undifferentiated).

**Figure 9 diseases-12-00111-f009:**
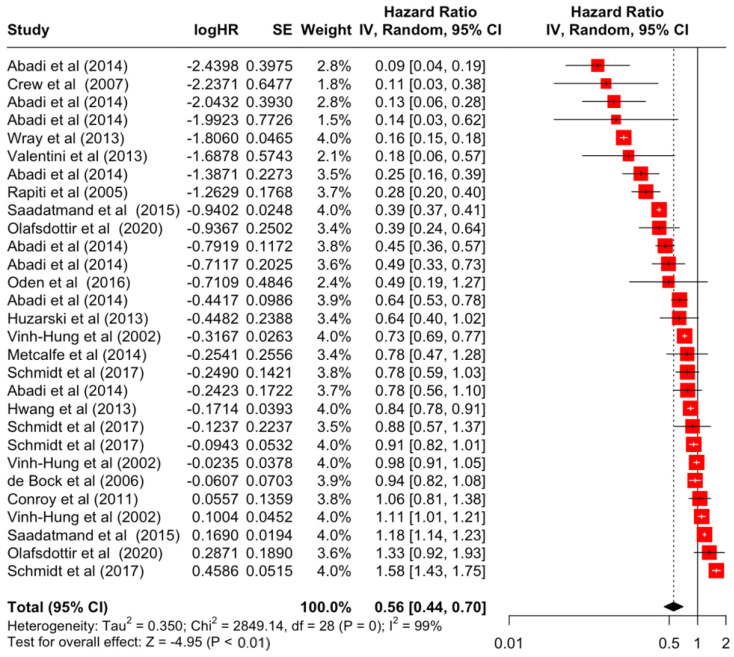
Forest Plot for Random-effects Hazard Ratio Model of Surgery (Yes).

**Figure 10 diseases-12-00111-f010:**
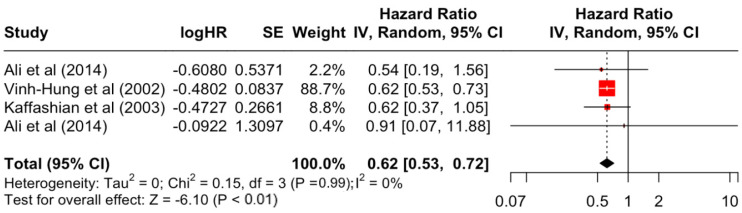
Forest Plot for Random-effects Hazard Ratio Model of Histology (Medullary).

**Figure 11 diseases-12-00111-f011:**
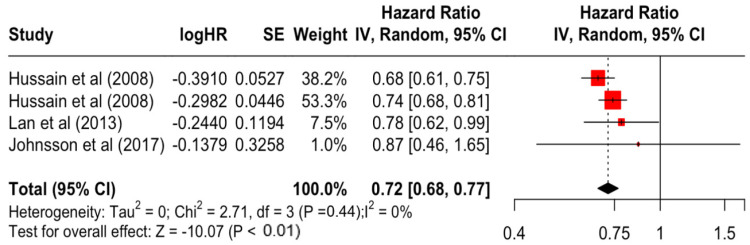
Forest Plot for Random-effects Hazard Ratio Model of Education (Higher).

**Figure 12 diseases-12-00111-f012:**
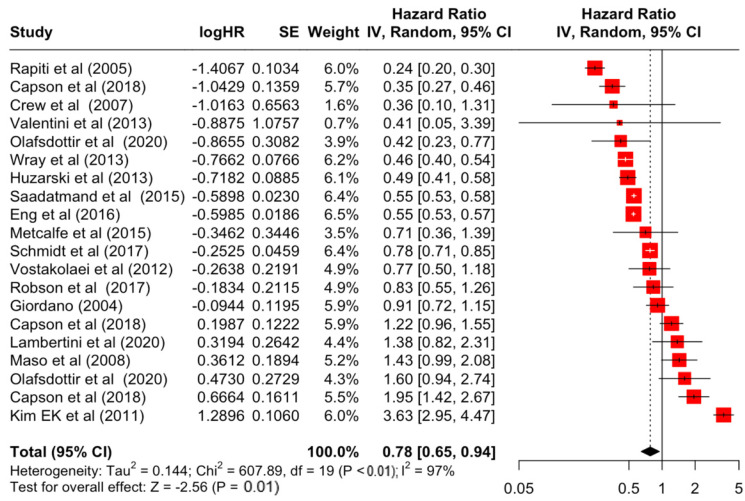
Forest Plot for Random-effects Hazard Ratio Model of Estrogen Receptor (Positive).

**Figure 13 diseases-12-00111-f013:**
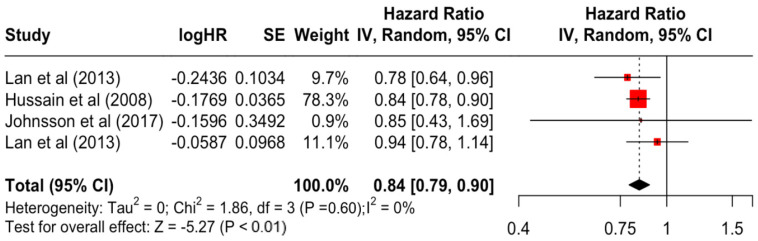
Forest Plot for Random-effects Hazard Ratio Model of Education (Secondary).

**Table 1 diseases-12-00111-t001:** Quality assessment according to a 6-component checklist from the Quality in Prognosis Studies (QUIPS) tool.

Study	Study Participation	Study Attrition	Prognostic Factor Measurement	Outcome Measurement	Study Confounding	Statistical Analysis and Reporting	Overall Quality
Rohan et al. [12]	x		x	x		x	Moderate
Watson et al. [13]	x		x	x		x	Moderate
Vinh-Hung et al. [14]	x		x	x	x	x	High
Kaffashian et al. [15]	x			x		x	Moderate
Robson et al. [16]	x	x	x	x		x	High
Metcalfe et al. [17]	x		x	x		x	Moderate
Boyapati [18]	x	x	x	x		x	High
Rapiti et al. [19]	x		x	x		x	Moderate
Le et al. [20]	x		x	x		x	Moderate
Field et al. [21]	x	x	x	x	x	x	High
de Bock et al. [22]	x		x	x		x	Moderate
Fink et al. [23].	x		x	x		x	Moderate
Sohn et al. [24]	x	x	x	x		x	High
Hussain et al. [25]	x	x	x	x		x	High
Maso et al. [26]	x		x	x	x	x	High
Nichols et al. [27]	x		x	x		x	Moderate
Gnerlich et al. [28]	x		x	x		x	Moderate
Peel et al. [29]	x	x	x	x		x	High
Powe et al. [30]	x		x	x	x	x	High
Bertram [31]	x		x	x	x	x	High
Ewertz et al. [32]	x	x	x	x		x	High
Conroy et al. [33]	x		x	x	x	x	High
Kim EK et al. [34]	x		x	x		x	Moderate
Colzani et al. [35]	x		x	x		x	Moderate
Goodwin et al. [36]	x		x	x	x	x	High
Lee et al. [37]	x		x	x		x	Moderate
Vostakolaei et al. [38]	x		x	x	x	x	High
Movahedi et al. [39]	x		x	x		x	Moderate
Lan et al. [40]	x		x	x	x	x	High
Keegan et al. [41]	x		x	x		x	Moderate
Hwang et al. [42]	x		x	x		x	Moderate
Nechuta et al. [43]	x		x	x		x	Moderate
Valentini et al. [44]	x		x	x		x	Moderate
Wray et al. [45]	x		x	x	x	x	High
Huzarski et al. [46]	x		x	x	x	x	High
Ali et al. [47]	x	x	x	x	x	x	High
de Glas et al. [48]	x		x	x		x	Moderate
Abadi et al. [49]	x		x	x		x	Moderate
Vollebergh et al. [50]	x		x	x		x	Moderate
Kriege et al. [51]	x		x	x	x	x	High
Saadatmand et al. [52]	x		x	x		x	Moderate
Kaplan et al. [53]	x		x	x	x	x	High
Eng et al. [54]	x		x	x		x	Moderate
Kim et al. [55]	x	x	x	x	x	x	High
Kataoka et al. [56]	x		x	x		x	Moderate
Partridge et al. [57]	x		x	x		x	Moderate
Odén et al. [58]	x	x	x	x		x	High
Chagpar et al. [59]	x		x	x		x	Moderate
Johnsson et al. [60]	x		x	x		x	Moderate
Fallahpour et al. [61]	x		x	x		x	Moderate
Schmidt et al. [62]	x		x	x		x	Moderate
Wong et al. [63]	x		x	x	x	x	High
Copson et al. [64]	x	x	x	x	x	x	High
Lambertini et al. [65]	x		x	x	x	x	High
Olafsdottir et al. [66]	x		x	x		x	Moderate
Talhouet et al. [67]	x		x	x		x	Moderate
Kim et al. [68]	x		x	x		x	Moderate

**Table 2 diseases-12-00111-t002:** Characteristics of individual selected studies.

Author	Year	Country	Study Design/Database	Sample Size	Sample	Prognostic Factor(s) Included
Rohan et al. [12]	1995	Australia	Population-based cohort study	412	Women with breast cancer	Physical activity level
Watson et al. [13]	1999	United Kingdom	Prospective survival study	578	Women with early-stage breast cancer	Mental Adjustment to Cancer (MAC) scale-predominant responses
Vinh-Hung et al. [14]	2002	USA	SEER 9-registries database	186,549	Women with partial or total mastectomy Breast cancer	Race, marital status, histology, differentiation, ER status, PR status, and treatment
Kaffashian et al. [15]	2003	United Kingdom	East Anglia Cancer Registry database	10,865	Women with breast cancer	Stage, grade, morphology, and social class
Robson et al. [16]	2003	USA	Retrospective cohort study	496	Women with breast cancer	BRCA1 mutation, tumour size, axillary node, and age of diagnosis
Metcalfe et al. [17]	2004	Canada	Retrospective cohort study	491	Women with breast cancer with BRCA1/2 mutation	BRCA
Boyapati [18]	2005	China	Retrospective cohort study	1459	Women with breast cancer	Total Isoflavone
Rapiti et al. [19]	2005	Switzerland	Retrospective data analysis	2997	Women with breast cancer	Age at diagnosis, method of discovery, socio-economic status, stage, histology, differentiation, ER status, surgery, radiotherapy, chemotherapy, and hormonal therapy
Le et al. [20]	2005	USA	Surveillance, Epidemiology and End Results Breast Implant Surveillance Study	4968	Women < 65 years with breast cancer	Implant status, age at diagnosis, race, grade, morphology, and radiation therapy
Field et al. [21]	2005	USA	Retrospective cohort study; Cancer research network	21,155	Women with breast cancer	Race, stage, grade, estrogen receptor, progesterone receptor, and tumour size
de Bock et al. [22]	2006	Netherlands	Retrospective data analysis	1073	Women with breast cancer	Age at diagnosis, tumour size, nodal state, surgical therapy, chemotherapy, adjuvant chemotherapy, adjuvant radiotherapy, and tamoxifen
Fink et al. [23]	2007	USA	Population-based case–control; Long Island Breast Cancer Study	1383	Women with breast cancer	Total Isoflavone
Sohn et al. [24]	2008	USA	Retrospective cohort study	13,984	Women with breast cancer	Age at diagnosis, race, grade, stage, and location
Hussain et al. [25]	2008	Sweden	Swedish Family Cancer Database	43,222	Women with invasive breast cancer	Histology, age at diagnosis, and education
Maso et al. [26]	2008	Italy	Multicentre case–control study	1453	Women with breast cancer	Age at diagnosis, tumour size, lymph node-positive, stage, ER status, PR status, BMI, work physical activity, leisure time, vegetable and fruit intake, total protein intake, total fat intake, and glycaemic load
Nichols et al. [27]	2009	USA	Population-based case–control	3993	Women with invasive non-metastatic breast cancers	BMI
Gnerlich et al. [28]	2009	USA	SEER 9-registries database	243,012	Women with breast cancer	Stage
Peel et al. [29]	2009	USA	Prospective study; Aerobics Center Longitudinal study	14,811	Women with breast cancer	Age at diagnosis, BMI, oral contraceptive use, and estrogen use
Powe et al. [30]	2010	United Kingdom	Prospective study	466	Women with breast cancer	Tumour size, grade, stage, and beta-blocker treatment
Bertram [31]	2011	USA	Randomized controlled trial	2361	Women with post-treatment breast cancer survivor	Total PA
Ewertz et al. [32]	2011	Denmark	Retrospective cohort study	5868	Women with early-stage breast cancer	BMI
Conroy et al. [33]	2011	USA	Prospective study (Multi-ethnic cohort study)	3842	Women with breast cancer aged 50 and above	Age at diagnosis, ethnicity, BMI, cardiovascular comorbidity, surgery, chemotherapy, and radiotherapy
Kim EK et al. [34]	2011	Korea	Nationwide registry from Seoul National University Hospital Breast Cancer Center (SNUHBCC) and Korean Breast Cancer Registry (KBCR)	2474	Women with breast cancer	Age at diagnosis, tumour size, LN positive, histology grade, hormone receptor, and HER2 status
Colzani et al. [35]	2011	Sweden	Stockholm Breast Cancer Registry	12,850	Women with breast cancer	Age at diagnosis, treatment, nodes, Estrogen-receptor status, and tumour size
Goodwin et al. [36]	2011	Canada, USA, Australia	International population-based cohort study	3220	Women with breast cancer	BRCA, chemotherapy, and hormone therapy
Lee et al. [37]	2011	USA	Retrospective data analysis; Clinical database and annotated Specialized Program of Research Excellence (SPORE)	117	Women with breast cancer (BRCA1 and noncarriers)	BRCA, Age, AJCC stage, lymph node, and tumour size
Vostakolaei et al. [38]	2012	Iran	Retrospective data analysis	1500	Women with breast cancer	Age at diagnosis, stage, grade, estrogen receptor, progesterone receptor, and HER2
Movahedi et al. [39]	2012	Iran	Retrospective data analysis	6147	Women with breast cancer	Age at diagnosis
Lan et al. [40]	2013	Vietnam	Retrospective data analysis	948	Women with breast cancer	Marital status, hormone therapy, education level, and stage
Keegan et al. [41]	2013	USA	California cancer registry	5331	Adolescent and young adult breast cancer	Her2, race, marital status, lymph nodes, and tumour grade
Hwang et al. [42]	2013	USA	Retrospective data analysis	112,514	Women with early-stage breast cancer	Surgery, grade, nodes, race, socio-economic status, tumour size, age at diagnosis, and hormone receptor status
Nechuta et al. [43]	2013	China	Population-based prospective study; Shanghai Breast Cancer Survival study	4664	Women with breast cancer	Comorbidity
Valentini et al. [44]	2013	Canada, USA, Asia, Europe	Multicentre, historical cohort study	397	Women with breast cancer (at least one mutation in the BRCA1 or BRCA2 gene)	Birth after diagnosis, age at diagnosis, chemotherapy, surgery, tumour size, lymph node, and receptor status
Wray et al. [45]	2013	USA	Retrospective data analysis; Harris Country Hospital District and Memorial Hermann Healthcare System	9249	Women with breast cancer	Age at diagnosis, race, stage, receptor, and hospital system
Huzarski et al. [46]	2013	Poland	Retrospective data analysis	3345	Women with breast cancer	Age at diagnosis, ER status, PR status, HER2 status, tumour size, nodes, oophorectomy, tamoxifen, chemotherapy, and BRCA
Ali et al. [47]	2014	Canada	Retrospective observational studies	8775	Women with breast cancer (estrogen-receptor positive)	Age at diagnosis, No. positive nodes, tumour size, grade, hormone therapy, chemotherapy, morphology, PR status, HER2 status, and molecular subtype
de Glas et al. [48]	2014	Netherlands	Randomised controlled trial	300	Women with breast cancer, aged < 65 at diagnosis, postmenopausal, hormone receptor-positive	Physical activity (MET-hrs/week)
Abadi et al. [49]	2014	Canada	Population-based British Columbia Cancer Registry	15,830	Women with breast cancer (Stage I ≤ 50 years)	Surgery
Vollebergh et al. [50]	2014	Netherlands	Retrospective multicentre RCT study	249	Women with breast cancer	Stage, grade, BRCA
Kriege et al. [51]	2014	Netherlands	Retrospective cohort study	4722	Women with breast cancer	Chemotherapy
Saadatmand et al. [52]	2015	Netherlands	Prospective nationwide population-based study	173,797	Women with breast cancer	Age at diagnosis, tumour category, pathological node category, morphology, Estrogen receptor status, Progesterone receptor status, HER2 status, breast surgery, chemotherapy, hormone therapy, and radiotherapy
Kaplan et al. [53]	2015	USA	Institutional Breast Cancer Clinical Database Registry	2998	Women with breast cancer aged 50 to 69	Detection method, radiation therapy, hormone treatment, and chemotherapy
Eng et al. [54]	2016	USA	SEER 18-registries database	25,323	Women with breast cancer (Stage IV)	Income status, age at diagnosis, tumour size, node status, estrogen receptor status, progesterone receptor status, and race
Kim et al. [55]	2016	USA	Population-based Long Islan Breast Cancer Study	1413	Women with breast cancer	Chemotherapy, hormone therapy, and radiation therapy
Kataoka et al. [56]	2016	Japan	Japanese Breast Cancer Registry	53,670	Women with breast cancer	Age at diagnosis, grade, node, subtype, and adjuvant therapy
Partridge et al. [57]	2016	USA	Longitudinal cohort study; National Comprehensive Cancer Network Breast Cancer Outcome Project database	17,575	Women with breast cancer	Age at diagnosis
Odén et al. [58]	2016	Canada	Prospective randomised trial	206	Women with breast cancer (BRCA1 or 2 mutation carriers)	BRCA, oophorectomy, and oral contraceptive use
Chagpar et al. [59]	2017	USA	Retrospective data analysis	157,584	Women aged ≥ 70 years diagnosed with cLN- HR+ breast cancer	Age at diagnosis, tumour size, race, tumour grade, and radiation therapy
Johnsson et al. [60]	2017	Sweden	Prospective population-based cohort; Swedish National Cancer Registry	847	Women with breast cancer	Stage, physical activity, oral contraception use, age at first childbirth, family history of breast cancer, education, BMI, smoking status, and alcohol
Fallahpour et al. [61]	2017	Canada	Population-based study; Ontario, Cancer Registry	17,598	Women with breast cancer (Luminal A)	Age at diagnosis, residence, histology, stage, and Charlson comorbidity index
Schmidt et al. [62]	2017	Netherlands	Retrospective cohort study	6478	Women with breast cancer (less than 50 years old)	BRCA, age at diagnosis, grade, tumour size, nodes, ER status, chemotherapy, and surgery
Wong et al. [63]	2018	Singapore	Retrospective data analysis; National Cancer Centre Singapore	2492	Women with breast cancer	Age at diagnosis
Copson et al. [64]	2018	United Kingdom	Prospective cohort study	2733	Women with breast cancer (40 years old or younger)	BRCA, BMI, grade, HER2 status, ER status, Race, and chemotherapy
Lambertini et al. [65]	2020	Europe, North America, Latin America, Israel	International, multicentre, retrospective cohort study	1252	Women with breast cancer (with germline deleterious BRCA mutations)	BRCA and hormone receptor status
Olafsdottir et al. [66]	2020	Nordics—Denmark, Iceland, Norway, Sweden	Retrospective data analysis	608	Women with breast cancer	Tumour size, lymph node, grade, ER status, surgery, chemotherapy, and radiation
Talhouet et al. [67]	2020	France, Switzerland	Retrospective cohort study	677 (French), 248 (Swiss)	Women with breast cancer (BRCA 1/2 or noncarriers)	BRCA, grade, age at diagnosis, and nodal status
Kim et al. [68]	2022	USA	SEER 18-registries database	158,253	Women with breast cancer (hormone receptor+, lower grade)	Age at diagnosis

**Table 3 diseases-12-00111-t003:** Pooled effects of breast cancer-specific survival by each prognostic factor.

Factor	Group	HR	Lower	Upper	z-Stats	*p*-Value
Age	Below 35	1.53	1.26	1.87	4.19	**<0.001**
	35 to 60	1.11	1.01	1.21	2.27	**0.023**
	Above 60	1.45	1.21	1.72	4.11	**<0.001**
Education	Secondary	0.84	0.79	0.90	−5.27	**<0.001**
	Higher	0.72	0.68	0.77	−10.07	**<0.001**
Race	Black	1.39	1.33	1.45	14.45	**<0.001**
	Asian	0.84	0.76	0.93	−3.23	**<0.001**
	Hispanic	1.16	0.98	1.36	1.74	0.082
Grade	2	1.54	1.27	1.87	4.44	**<0.001**
	3	1.92	1.33	2.76	3.47	**<0.001**
Stage	2	1.93	1.48	2.51	4.86	**<0.001**
	3	3.42	2.51	4.67	7.76	**<0.001**
	4	12.12	5.70	25.76	6.48	**<0.001**
Differentiation	Moderate	1.49	1.15	1.93	3.02	**0.003**
	Poor	2.43	1.79	3.30	5.72	**<0.001**
	Undifferentiated	2.24	1.66	3.01	5.3	**<0.001**
Nodes	Positive	1.71	1.42	2.05	5.77	**<0.001**
Surgery	Yes	0.56	0.44	0.70	−4.95	**<0.001**
Tumour size	≥2 cm	1.39	1.35	1.42	24.75	**<0.001**
Histology	Lobular	1.09	0.88	1.34	0.81	0.420
	Medullary	0.62	0.53	0.72	−6.1	**<0.001**
	Others	0.90	0.71	1.15	−0.81	0.420
Estrogen receptor	Positive	0.78	0.65	0.94	−2.56	**0.011**
	Negative	1.55	1.13	2.13	2.69	**0.007**
HER2 receptor	Positive	1.29	0.93	1.79	1.53	0.125
	Negative	1.16	0.79	1.70	0.76	0.448
Body Mass Index	Overweight/Obese	1.20	1.09	1.33	3.69	**<0.001**
Comorbidity Index	1 to 2	1.87	1.35	2.61	3.72	**<0.001**
	≥3	3.29	2.40	4.52	7.35	**<0.001**
BRCA	1	0.86	0.61	1.19	−0.91	0.363
	2	1.02	0.78	1.33	0.13	0.898
Oral contraceptive use	Yes	0.97	0.72	1.31	−0.19	0.852
Physical activity	Light/Moderate	0.98	0.80	1.20	−0.21	0.833
	High/Vigorous	0.99	0.68	1.44	−0.06	0.950
Progesterone receptor	Positive	0.87	0.70	1.07	−1.3	0.193
	Negative	1.21	0.78	1.88	0.86	0.391
Hormone therapy	Yes	0.98	0.74	1.29	−0.16	0.872
Chemotherapy	Yes	1.02	0.81	1.28	0.16	0.869
Radiotherapy	Yes	1.14	0.83	1.57	0.8	0.423
Tamoxifen	Yes	0.74	0.48	1.16	−1.32	0.187
**Bold values indicate statistical significance at 0.05**						

**Table 4 diseases-12-00111-t004:** Between-study heterogeneity statistics of each prognostic factor.

Factor	Group	Q-Statistics	df	*p*-Value	I-Square	Classification
Age	Below 35	8.81	4	0.066	55%	moderate
	35 to 60	629.27	41	**<0.001**	93%	very high
	Above 60	731.92	30	**<0.001**	96%	very high
Education	Secondary	1.86	3	0.601	0%	low
	Higher	2.71	3	0.439	0%	low
Race	Black	17.21	11	0.102	36%	moderate
	Asian	9.78	6	0.134	39%	moderate
	Hispanic	22.9	5	**<0.001**	78%	very high
Grade	2	164.76	11	**<0.001**	93%	very high
	3	1077.9	12	**<0.001**	99%	very high
Stage	2	352.18	13	**<0.001**	96%	very high
	3	659	16	**<0.001**	98%	very high
	4	4008.6	13	**<0.001**	100%	very high
Differentiation	Moderate	21.59	3	**<0.001**	86%	very high
	Poor	35.59	4	**<0.001**	89%	very high
	Undifferentiated	61.11	6	**<0.001**	90%	very high
Nodes	Positive	2295.3	32	**<0.001**	99%	very high
Surgery	Yes	2849.14	28	**<0.001**	99%	very high
Tumour size	≥2 cm	2409.95	24	**<0.001**	99%	very high
Histology	Lobular	170.94	11	**<0.001**	94%	very high
	Medullary	0.15	3	0.986	0%	low
	Others	403.92	12	**<0.001**	97%	very high
Estrogen receptor	Positive	607.89	19	**<0.001**	97%	very high
	Negative	244.5	6	**<0.001**	98%	very high
HER2 receptor	Positive	470.86	10	**<0.001**	98%	very high
	Negative	23.44	2	**<0.001**	91%	very high
Body Mass Index	Overweight/Obese	36.79	11	**<0.001**	70%	high
Comorbidity Index	1 to 2	6.36	5	0.273	21%	low
	≥3	9.64	3	**0.022**	69%	high
BRCA	1	45.2	8	**<0.001**	82%	very high
	2	27.12	9	**<0.001**	67%	high
Oral contraceptive use	Yes	1.83	2	0.400	0%	low
Physical activity	Light/Moderate	3.63	4	0.459	0%	low
	High/Vigorous	6.1	5	0.296	18%	low
Progesterone receptor	Positive	468.94	12	**<0.001**	97%	very high
	Negative	536.84	6	**<0.001**	99%	very high
Hormone therapy	Yes	180.85	10	**<0.001**	94%	very high
Chemotherapy	Yes	935.86	27	**<0.001**	97%	very high
Radiotherapy	Yes	1329.58	16	**<0.001**	99%	very high
Tamoxifen	Yes	13.95	2	**<0.001**	86%	very high
Bold values indicate statistical significance at 0.05						

**Table 5 diseases-12-00111-t005:** Egger’s test results for publication bias.

Factor	Group	t-Statistics	df	*p*-Value
Age	Below 35	−1.14	3	0.336
	35 to 60	2.39	40	**0.022**
	Above 60	−1.51	29	0.141
Education	Secondary	0.26	2	0.822
	Higher	0.72	2	0.546
Race	Black	−0.16	10	0.873
	Asian	−0.07	5	0.943
	Hispanic	1.33	4	0.256
Grade	2	−0.27	10	0.793
	3	−0.41	11	0.692
Stage	2	1.62	12	0.132
	3	2.12	15	**0.052**
	4	−0.57	12	0.577
Differentiation	Moderate	2.62	2	0.120
	Poor	2.13	3	0.123
	Undifferentiated	3.09	5	**0.027**
Nodes	Positive	0.81	31	0.427
Surgery	Yes	−0.89	27	0.383
Tumour size	≥2 cm	5.96	23	**<0.001**
Histology	Lobular	0.03	10	0.978
	Medullary	0.27	2	0.812
	Others	−1.32	11	0.215
Estrogen receptor	Positive	1.58	18	0.132
	Negative	−0.76	5	0.479
HER2 receptor	Positive	−0.03	9	0.978
	Negative	0.49	1	0.713
Body Mass Index	Overweight/Obese	3.38	10	**0.007**
Comorbidity Index	1 to 2	0.45	4	0.678
	≥3	0.57	2	0.628
BRCA	1	−1.07	7	0.321
	2	−0.29	8	0.778
Oral contraceptive use	Yes	0.96	1	0.513
Physical activity	Light/Moderate	−1.22	3	0.309
	High/Vigorous	−1.1	4	0.335
Progesterone receptor	Positive	1.67	11	0.123
	Negative	−0.83	5	0.447
Hormone therapy	Yes	−0.95	9	0.368
Chemotherapy	Yes	2.79	26	**0.010**
Radiotherapy	Yes	3.56	15	**0.003**
Tamoxifen	Yes	−0.04	1	0.972
Bold values indicate statistical significance at 0.05				

**Table 6 diseases-12-00111-t006:** Meta-regression univariable analysis for the effect of each prognostic factor association with the year of study and sample size.

		Year of Study		
		Estimate	SE	*p*-Value
Age	Below 35	−0.01	0.02	0.598
	35 to 60	0.01	0.01	0.199
	Above 60	0.05	0.01	**0.002**
Education	Secondary	0.01	0.02	0.699
	Higher	0.02	0.02	0.344
Race	Black	0.00	0.00	0.706
	Asian	0.01	0.02	0.578
	Hispanic	0.00	0.03	0.853
Grade	2	0.00	0.02	0.961
	3	−0.02	0.03	0.661
Stage	2	−0.06	0.03	0.061
	3	0.02	0.04	0.533
	4	0.09	0.06	0.108
Differentiation	Moderate	0.19	0.29	0.519
	Poor	0.09	0.22	0.677
	Undifferentiated	0.14	0.16	0.394
Nodes	Positive	−0.01	0.03	0.646
Surgery	Yes	0.00	0.03	0.911
Tumour size	≥2 cm	0.02	0.01	0.145
Histology	Lobular	−0.02	0.11	0.872
	Medullary	0.00	0.04	0.983
	Others	−0.01	0.04	0.898
Estrogen receptor	Positive	−0.03	0.08	0.747
	Negative	0.00	0.02	0.944
HER2 receptor	Positive	0.00	0.04	0.918
	Negative	−0.05	0.03	**0.036**
Body Mass Index	Overweight/Obese	−0.01	0.03	0.583
Comorbidity Index	1 to 2	−0.02	0.05	0.682
	≥3	0.02	0.02	0.314
BRCA	1	−0.02	0.02	0.386
	2	0.00	0.09	0.977
Oral contraceptive use	Yes	−0.34	0.19	0.073
Physical activity	Light/Moderate	0.01	0.02	0.453
	High/Vigorous	−0.05	0.03	0.088
Progesterone receptor	Positive	0.01	0.02	0.531
	Negative	−0.06	0.03	0.052
Hormone therapy	Yes	0.00	0.03	0.968
Chemotherapy	Yes	0.06	0.04	0.181
Radiotherapy	Yes	−0.04	0.04	0.365
Tamoxifen	Yes	-	-	-
Bold values indicate statistical significance at 0.05				

## Data Availability

The data that support the findings of this study are not openly available due to institutional permission but are available from the corresponding author upon reasonable request.

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
