# Peer review of "Prognostic Factors Associated with Breast Cancer-Specific Survival from 1995 to 2022: A Systematic Review and Meta-Analysis of 1,386,663 Cases from 30 Countries"

_diseases, 2024, doi:10.3390/diseases12060111_

Round 1
Reviewer 1 Report
Comments and Suggestions for Authors
The manuscript is very well written, although it should be re-edited because it is completely unreadable due to the huge number of figures and tables - some of these materials should be included in the supplement to the work and it should be completely re-edited. When it comes to statistics, it is carried out as correctly as possible and the manuscript looks completely correct - the discussion should be broader, discussing the results obtained by the authors.
Author Response
Dear reviewer,
Thank you for your valuable input of our humble manuscript. We agree that the number of Figures are too much and have moved them into a Supplementary file. Only the Top-5 important factors are retained in the Figures.
We greatly appreciate your feedback and hope for your favorable decision soon.
Wishing you good health and peace.

Reviewer 2 Report
Comments and Suggestions for Authors
This meta-analysis estimated the pooled effects of breast cancer-specific survival in a large sample originated from 30 countries.
The paper is comprehensive, but there are still shortcomings that need to be revised:
1. The methodological description in the paper is too simplistic.
2. The discussion section is too simplistic and requires a detailed analysis of the Top-5 best prognostic factor.
3. Did the author consider the differences in treatment levels between different countries and regions, as well as between different years, in this study.
Author Response
Dear reviewer,
Thank you for your valuable input. Kindly find below point-by-point responses to your comments.
- The methodological description in the paper is too simplistic. Response: Step-by-step approach following PRISMA guideline was conducted for this systematic review, further details added.
- The discussion section is too simplistic and requires a detailed analysis of the Top-5 best prognostic factor. Response: Further discussion was added.
- Did the author consider the differences in treatment levels between different countries and regions, as well as between different years, in this study. Response: There was limited information available to extract on treatment level, as well as survival rate but where is available, the rates vary greatly from 1 to 15 years - because there is no standardized rate, this is reported as limitations of this review.
We greatly appreciate your feedback and hope to receive your favorable decision soon. Thank you and have a a pleasant day.

Reviewer 3 Report
Comments and Suggestions for Authors
This review investigates the possible prognostic factors of the breast cancer survival include 101 studies. 13 out of 22 prognostic factors were in terms of the breast cancer survival. Top-5 poorest prognostic factors were stage 4, stage 3, comorbidity index ≥3, poor differentiation of cancer cell histology, and undifferentiated cancer cell histology. Top-5 best prognostic factors different types of mastectomy and breast conserving therapies, medullary histology, higher education, and positive estrogen receptor status.
I just have a few questions:
In the table 2, you mentioned there are 2 papers describe the men with breast cancer, what are the main factors for the survival in men? What are the different factors that affect male and female survival?
As mentioned in the Discussion section, these studies used survival rates associated with different years of survival. Have you normalized this? How do you compare the differences across all studies?
Author Response
Dear reviewer,
Thank you for your valuable input. Kindly find point-by-point response to your comments/questions below:
In the table 2, you mentioned there are 2 papers describe the men with breast cancer, what are the main factors for the survival in men? What are the different factors that affect male and female survival?
Response: The review only include Female patients and the two reports were actually excluded from the analysis. We have corrected Table 2.
As mentioned in the Discussion section, these studies used survival rates associated with different years of survival. Have you normalized this? How do you compare the differences across all studies?
Response: The pooled effects from meta-analysis adjusted for variability. Nevertheless, we acknowledge that this is a limitation as each studies may report different survival rate.
We greatly appreciate your feedback. Hope to receive your favorable decision soon. Thank you and have a pleasant day.

Reviewer 4 Report
Comments and Suggestions for Authors
Thank you for submitting this interesting review however although it included a large number of cases and from different countries, the impact of the study was very limited. The study lacked novelty that could add to our knowledge and to the literatures. There were so many limitations of the study to make any meaningful interpretation and to make any solid conclusion. Many studies included in the review had been published by the either same first authors or one of the coauthors.
specific comments:
1- Thank you for submitting this interesting meta analysis addressing the prognostic factors for breast cancer specific survival . The analysis included a large number of articles and cases over more than 10 years in many countries however there were so many limitations for such analysis that could lead to misinterpretation of results and flawed conclusions for instance
1.1 No control for many confounders including for instance the quality of health care and type of treatment received. In additions, given these studies run by different teams in different areas over an extended period of time, the differences in the methodology for evaluating each prognostic factors would be different too. One obvious example was the association between HER2 and survival and which scores have been adopted as in patients who treated before 2006 and did not received HER2 target therapy, the association would be negative while in patients who treated after 2006 and received Herceptin this association would be in the opposite direction.
1.2 The heterogeneity and inconsistency in patients population, treatment, health care and evaluation of each prognostic parameters. For examples the histological grading which reflects de-differentiation status which the authors included it as another separate factors. Also including age as a prognostic parameter rather than menopausal status which in turn related to the treatment that would be received specially in ER positive breast cancer. Moreover the inconsistent used for staging lymph nodes metastasis sometimes it was 3- ties vs 4-ties. In many studies the lymph nodes stage was just negative or positive. That is why the authors considered two prognostic factor for lymph nodes.
1.3 Some parameters were presented in small number of articles and tested in small number of patients.
1.4 Many articles included in the meta analysis were conducted by same team and coauthors.
2. The study suffered from lack of novelty and the study would not add to our knowledge or to the literature as the ER status, HER2 status, histological grade, lymph node stage and tumour size were used as well known prognostic factors in breast cancer for many years and adopted by clinicians for management of breast cancer for more than 20 years now. So the clinical utility or implications of such study is very limited as it validated what has already been validated for many years.
3- More than 250 new bio marks or genomic tests used to be published every year and the authors did not include any of these tests or marker in the meta analysis.
4- the meta analysis just included univariate analysis and did not include multivariable cox regression analysis or any other methods.
5- The conclusions were not consistent with the
evidence and arguments presented as it based on misinterpretation of the results and incorrect methods. In the figures and tables , individual studies showed different direction of association between survival and same parameters. That is why the pool analysis would give misleading results biased by weight or number of cases in each study in each direction.
6- The references were not appropriate as some of the references were published by the same team or coauthors.
7- Including a large number of heterogenous cases using inconsistent method of evaluation would generate poor quality of data, waste time and effort and misleading conclusions.
Comments on the Quality of English Language
The English language needs significant editing.
Author Response
Dear reviewer,
Thank you for your valuable input. Kindly find point-by-point response to your comments/questions below:
1.1 No control for many confounders including for instance the quality of health care and type of treatment received. In additions, given these studies run by different teams in different areas over an extended period of time, the differences in the methodology for evaluating each prognostic factors would be different too. One obvious example was the association between HER2 and survival and which scores have been adopted as in patients who treated before 2006 and did not received HER2 target therapy, the association would be negative while in patients who treated after 2006 and received Herceptin this association would be in the opposite direction.
Response: This is an important point for primary studies, we acknowledge that in any review, it will inherit the shortcomings of each papers. In this review, we ensured to use one standardized measure - Hazard ratio, that is adjusted across time, studies, etc.
1.2 The heterogeneity and inconsistency in patients population, treatment, health care and evaluation of each prognostic parameters. For examples the histological grading which reflects de-differentiation status which the authors included it as another separate factors. Also including age as a prognostic parameter rather than menopausal status which in turn related to the treatment that would be received specially in ER positive breast cancer. Moreover the inconsistent used for staging lymph nodes metastasis sometimes it was 3- ties vs 4-ties. In many studies the lymph nodes stage was just negative or positive. That is why the authors considered two prognostic factor for lymph nodes.
Response: The factors are extracted as reported in the included studies. In meta-analysis, the comparison cuts across the results of different categories from different studies i.e., different study population, independently. It is different from primary studies that analyses each factor within its subgroups, which makes sense as they are from the same study population.
1.3 Some parameters were presented in small number of articles and tested in small number of patients.
Response: Yes, this is an unavoidable limitation of systematic review, where the analysis is based on availability of eligible reports.
1.4 Many articles included in the meta analysis were conducted by same team and coauthors.
Response: It makes sense that the same field experts published several papers within the time span of this review that attempts to capture the factors within the last 20 years. Where two data are extracted from the same paper is due to two different study populations were reported.
- The study suffered from lack of novelty and the study would not add to our knowledge or to the literature as the ER status, HER2 status, histological grade, lymph node stage and tumour size were used as well known prognostic factors in breast cancer for many years and adopted by clinicians for management of breast cancer for more than 20 years now. So the clinical utility or implications of such study is very limited as it validated what has already been validated for many years.
Response: It is obvious that these results are not new, as data is extracted from articles already published. The primary aim of systematic review and meta-analysis is to provide accumulative evidence of these published results that by all means is not always possible to be conclusive as research is an endless endeavor and more data will be generated over time. This review provides, at this juncture, the pooled, summative results of the evidence thus far, for stakeholders consideration.
3- More than 250 new bio marks or genomic tests used to be published every year and the authors did not include any of these tests or marker in the meta analysis.
Response: We totally agree and had an extensive discussion about this area at the beginning stages. As this review focuses on breast-cancer specific survival, we have to omit the studies we found that uses genetic markers that uses all-cause survival, among other reasons such as prediction only, unadjusted HR, etc. and therefore we end up with only clinical markers after filtering for eligible papers.
4- the meta analysis just included univariate analysis and did not include multivariable cox regression analysis or any other methods.
Response: We believe you are referring to Kaplan Meier survival analysis and Multivariable cox proportionate hazard regression analysis, where these are analysis used for primary data and not appropriate in systematic review and meta-analysis. In our meta-analysis, we have further conducted between-study heterogeneity and meta-regression to further validate the findings.
5- The conclusions were not consistent with the evidence and arguments presented as it based on misinterpretation of the results and incorrect methods. In the figures and tables , individual studies showed different direction of association between survival and same parameters. That is why the pool analysis would give misleading results biased by weight or number of cases in each study in each direction.
Response: Kindly see response for point 1.2 and 4.
6- The references were not appropriate as some of the references were published by the same team or coauthors.
Response: Kindly see point 1.4.
7- Including a large number of heterogenous cases using inconsistent method of evaluation would generate poor quality of data, waste time and effort and misleading conclusions.
Response: We would appreciate more specific explanation on this point. In any systematic review and meta-analysis, heterogeneity is unavoidable as data are extracted from published articles from different sources, countries, study sample, etc. We had followed the PRISMA guidelines and utilises widely used methods of evaluation such as quality assessment of individual studies, Q-statistics and I-square statistics for heterogeneity, Egger's test and funnel plot for publication bias, and meta-regression. Furthermore, as mentioned above in point 2, the findings are by no means conclusive as this review provides, at this juncture, the pooled, summative results of the evidence within the review scope, thus far, for stakeholders consideration.
We greatly appreciate your feedback. Hope to hear favorable decision from you soon. Thank you and have a pleasant day ahead.

Round 2
Reviewer 1 Report
Comments and Suggestions for Authors
The authors significantly improved the entire manuscript and implemented all comments - congratulations on a very good work. I have no further comments.
Reviewer 4 Report
Comments and Suggestions for Authors
Thank you for submitting this revised review which has addressed my concerns significantly.
Comments on the Quality of English LanguageNo major issues